# WaveCorr: Deep Reinforcement Learning with Permutation Invariant Policy Networks for Portfolio Management

## Abstract

The problem of portfolio management represents an important and challenging class of dynamic decision making problems, where rebalancing decisions need to be made over time with the consideration of many factors such as investors' preferences, trading environment, and market conditions. In this paper, we present a new portfolio policy network architecture for deep reinforcement learning (DRL) that can exploit more effectively cross-asset dependency information and achieve better performance than state-of-the-art architectures. In doing so, we introduce a new form of permutation invariance property for policy networks and derive general theory for verifying its applicability. Our portfolio policy network, named WaveCorr, is the first convolutional neural network architecture that preserves this invariance property when treating asset correlation information. Finally, in a set of experiments conducted using data from both Canadian (TSX) and American stock markets (S&P 500), WaveCorr consistently outperforms other architectures with an impressive 3%-25% absolute improvement in terms of average annual return, and up to more than 200% relative improvement in average Sharpe ratio. We also measured an improvement of a factor of up to 5 in the stability of performance under random choices of initial asset ordering and weights. The stability of the network has been found as particularly valuable by our industrial partner.

## 1 Introduction

In recent years, there has been a growing interest in applying Deep Reinforcement Learning (DRL) to solve dynamic decision problems that are complex in nature. One representative class of problems is portfolio management, whose formulation typically requires a large amount of continuous state/action variables and a sophisticated form of risk function for capturing the intrinsic complexity of financial markets, trading environment, and investors' preferences. In this paper, we propose a new architecture of DRL for solving portfolio management problems that optimize a Sharpe ratio criteria. While there are several works in the literature that apply DRL for portfolio management problems such as Moody et al. (1998); He et al. (2016); Liang et al. (2018) among others, little has been done to investigate how to improve the design of a Neural Network (NN) in DRL so that it can capture more effectively the nature of dependency exhibited in financial data. In particular, it is known that extracting and exploiting cross-asset dependencies over time is crucial to the performance of portfolio management. The neural network architectures adopted in most existing works, such as Long-Short-Term-Memory (LSTM) or Convolutional Neutral Network (CNN), however, only process input data on an asset-by-asset basis and thus lack a mechanism to capture cross-asset dependency information. The architecture presented in this paper, named as WaveCorr, offers a mechanism to extract the information of both time-series dependency and cross-asset dependency. It is built upon the WaveNet structure (Oord et al., 2016), which uses dilated causal convolutions at its core, and a new design of correlation block that can process and extract cross-asset information.

In particular, throughout our development, we identify and define a property that can be used to guide the design of a network architecture that takes multi-asset data as input. This property, referred to as *asset permutation invariance*, is motivated by the observation that the dependency across assets has a very different nature from the dependency across time. Namely, while the dependency across time is sensitive to the sequential relationship of data, the dependency across assets is not. To put

it another way, given a multivariate time series data, the data would not be considered the same if the time indices are permuted, but the data should remain the same if the asset indices are permuted. While this property may appear more than reasonable, as discussed in section 3, a naive extension of CNN that accounts for both time and asset dependencies can easily fail to satisfy this property. To the best of our knowledge, the only other works that have also considered extracting cross-asset dependency information in DRL for portfolio management are the recent works of Zhang et al. (2020) and Xu et al. (2020). While Zhang et al.'s work is closer to ours in that it is also built upon the idea of adding a correlation layer to a CNN-like module, its overall architecture is different from ours and, most noticeably, their design does not follow the property of asset permutation invariance and thus its performance can vary significantly when the ordering of assets changes. As further shown in the numerical section, our architecture, which has a simpler yet permutation invariant structure, outperforms in many aspects Zhang et al.'s architecture. The work of Xu et al. (2020) takes a very different direction from ours, which follows a so-called attention mechanism and an encoder-decoder structure. A more detailed discussion is beyond the scope of this paper.

Overall, the contribution of this paper is three fold. First, we introduce a new permutation invariance property for policy network architectures, referred to as asset permutation invariance in the case of a portfolio policy network, and derive general theory for verifying its applicability. Second, we design the first CNN based portfolio policy network, named WaveCorr, that accounts for asset dependencies in a way that preserves this type of invariance. This achievement relies on the design of an innovative permutation invariant correlation processing layer. Third, and most importantly, we present evidence that WaveCorr significantly outperforms state-of-the-art policy network architectures using data from both Canadian (TSX) and American (S&P 500) stock markets. Specifically, our new architecture leads to an impressive 5%-25% absolute improvement in terms of average annual return, up to more than 200% relative improvement in average Sharpe ratio, and reduces, during the period of 2019-2020 (i.e. the Covid-19 pandemic), by 16% the maximum daily portfolio loss compared to the best competing method. Using the same set of hyper-parameters, we also measured an improvement of up to a factor of 5 in the stability of performance under random choices of initial asset ordering and weights, and observe that WaveCorr consistently outperforms our benchmarks under a number of variations of the model: including the number of available assets, the size of transaction costs, etc. Overall, we interpret this empirical evidence as strong support regarding the potential impact of the WaveCorr architecture on automated portfolio management practices, and, more generally, regarding the claim that asset permutation invariance is an important NN property for this class of problems.

The rest of the paper unfolds as follows. Section 2 presents the portfolio management problem and risk averse reinforcement learning formulation. Section 3 introduces the new property of "asset permutation invariance" for portfolio policy network and presents a new network architecture based on convolution networks that satisfies this property. Finally, Section 4 presents the findings from our numerical experiments.

## 2 PROBLEM STATEMENT

### 2.1 PORTFOLIO MANAGEMENT PROBLEM

The portfolio management problem consists of optimizing the reallocation of wealth among many available financial assets including stocks, commodities, equities, currencies, etc. at discrete points in time. In this paper, we assume that there are $m$ risky assets in the market, hence the portfolio is controlled based on a set of weights $\boldsymbol{w}_t \in \mathbb{W} := \{\boldsymbol{w} \in \mathbb{R}^m_+ | \sum_{i=1}^m w^i = 1\}$, which describes the proportion of wealth invested in each asset. Portfolios are rebalanced at the beginning of each period $t = 0, 1, ..., T-1$, which will incur proportional transaction costs for the investor, i.e. commission rates are of $c_s$ and $c_p$, respectively. We follow Jiang et al. (2017) to model the evolution of the portfolio value and weights (see Figure 1). Specifically, during period $t$ the portfolio value and weights start at $p_{t-1}$ and $\boldsymbol{w}_{t-1}$, and the changes in stock prices, captured by a random vector of asset returns $\boldsymbol{\xi}_t \in \mathbb{R}^m$, affect the end of period portfolio value $p'_t := p_{t-1}\boldsymbol{\xi}_t^\top \boldsymbol{w}_{t-1}$, and weight vector $\boldsymbol{w}'_t := (p_{t-1}/p'_t)\boldsymbol{\xi}_t \bullet \boldsymbol{w}_{t-1}$, where $\bullet$ is a term-wise product. The investor then decides on a new distribution of his wealth $\boldsymbol{w}_t$, which triggers the following transaction cost:

$$c_s \sum_{i=1}^m (p'_t w'^i_t - p_t w^i_t)^+ + c_p \sum_{i=1}^m (p_t w^i_t - p'_t w'^i_t)^+ .$$

Denoting the net effect of transaction costs on portfolio value with $\nu_t := p_t/p'_t$, as reported in Li et al. (2018) one finds that $\nu_t$ is the solution of the following equations:

$$\nu_t = f(\nu_t, \boldsymbol{w}'_t, \boldsymbol{w}_t) := 1 - c_s \sum_{i=1}^{m} (w'^i_t - \nu_t w^i_t)^+ - c_p \sum_{i=1}^{m} (\nu_t w^i_t - w'^i_t)^+.$$

This, in turn, allows us to express the portfolio's log return during the $t+1$-th period as:

$$\zeta_{t+1} := \ln(p'_{t+1}/p'_t) = \ln(\nu_t p'_{t+1}/p_t) = \ln(\nu_t(\boldsymbol{w}'_t, \boldsymbol{w}_t)) + \ln(\boldsymbol{\xi}^\top_{t+1} \boldsymbol{w}_t) \qquad (1)$$

where we make explicit the influence of $\boldsymbol{w}'_t$ and $\boldsymbol{w}_t$ on $\nu_t$.

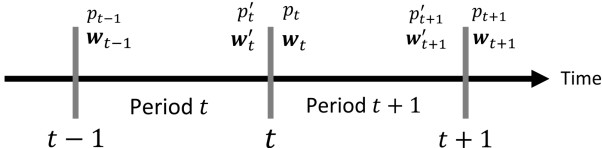

Figure 1: Portfolio evolution through time

We note that in Jiang et al. (2017), the authors suggest to approximate $\nu_t$ using an iterative procedure. However, we actually show in Appendix A.1 that $\nu_t$ can easily be identified with high precision using the bisection method.

## 2.2 RISK-AVERSE REINFORCEMENT LEARNING FORMULATION

In this section, we formulate the portfolio management problem as a Markov Decision Process (MDP) denoted by $(\mathcal{S}, \mathcal{A}, r, P)$. In this regard, the agent (i.e. an investor) interacts with a stochastic environment by taking an action $a_t \equiv \boldsymbol{w}_t \in \mathbb{W}$ after observing the state $s_t \in \mathcal{S}$ composed of a window of historical market observations, which include the latest stock returns $\boldsymbol{\xi}_t$, along with the final portfolio composition of the previous period $\boldsymbol{w}'_t$. This action results in the immediate stochastic reward that takes the shape of an approximation of the realized log return, i.e. $r_t(s_t, a_t, s_{t+1}) := \ln(f(1, \boldsymbol{w}'_t, \boldsymbol{w}_t)) + \ln(\boldsymbol{\xi}^\top_{t+1} \boldsymbol{w}_t) \approx \ln(\nu(\boldsymbol{w}'_t, \boldsymbol{w}_t)) + \ln(\boldsymbol{\xi}^\top_{t+1} \boldsymbol{w}_t)$, for which a derivative is easily obtained. Finally, $P$ captures the assumed Markovian transition dynamics of the stock market and its effect on portfolio weights: $P(s_{t+1}|s_0, a_0, s_1, a_1, ..., s_t, a_t) = P(s_{t+1}|s_t, a_t)$.

Following the works of Moody et al. (1998) and Almahdi & Yang (2017) on risk averse DRL, our objective is to identify a deterministic trading policy $\mu_\theta$ (parameterized by $\theta$) that maximizes the expected value of the Sharpe ratio measured on $T$-periods log return trajectories generated by $\mu_\theta$. Namely:

$$\max_\theta J_F(\mu_\theta) := \mathbb{E}_{\substack{s_0 \sim F \\ s_{t+1} \sim P(\cdot|s_t, \mu_\theta(s_t))}} [SR(r_0(s_0, \mu_\theta(s_0), s_1), ..., r_{T-1}(s_{T-1}, \mu_\theta(s_{T-1}), s_T))] \qquad (2)$$

where $F$ is some fixed distribution and

$$SR(r_{0:T-1}) := \frac{(1/T) \sum_{t=0}^{T-1} r_t}{\sqrt{(1/(T-1)) \sum_{t=0}^{T-1} (r_t - (1/T) \sum_{t=0}^{T-1} r_t)^2}}.$$

The choice of using the Sharpe ratio of log returns is motivated by modern portfolio theory (see Markowitz (1952)), which advocates a balance between expected returns and exposure to risks, and where it plays the role of a canonical way of exercising this trade-off (Sharpe, 1966). While it is inapt of characterizing downside risk, it is still considered a "gold standard of performance evaluation" by the financial community (Bailey & Lopez de Prado, 2012). In Moody et al. (1998), the trajectory-wise Sharpe ratio is used as an estimator of the instantaneous one in order to facilitate its use in RL. A side-benefit of this estimator is to offer some control on the variations in the evolution of the portfolio value which can be reassuring for the investor.

In the context of our portfolio management problem, since $s_t$ is composed of an exogeneous component $s_t^{exo}$ which includes $\boldsymbol{\xi}_t$ and an endogenous state $\boldsymbol{w}'_t$ that becomes deterministic when $a_t$ and $s_{t+1}^{exo}$ are known, we have that:

$$J_F(\mu_\theta) := \mathbb{E}_{\substack{s_0 \sim F \\ s_{t+1} \sim P(\cdot|s_t, \beta(s_t))}} [SR(r_0(\bar{s}^\theta_0, \mu_\theta(\bar{s}^\theta_0), \bar{s}^\theta_1), \ldots, r_{T-1}(\bar{s}^\theta_{T-1}, \mu_\theta(\bar{s}^\theta_{T-1}), \bar{s}^\theta_T))]$$

where $\beta(s_t)$ is an arbitrary policy, and where the effect of $\mu_\theta$ on the trajectory can be calculated using

$$\bar{s}_t^\theta := \left( s_t^{exo}, \frac{\boldsymbol{\xi}_t \bullet \mu_\theta(\bar{s}_{t-1}^\theta)}{\boldsymbol{\xi}_t^\top \mu_\theta(\bar{s}_{t-1}^\theta)} \right),$$

for $t \geq 1$, while $\bar{s}_0^\theta := s_0$. Hence,

$$\nabla_\theta J_F(\mu_\theta) := \mathbb{E}[\nabla_\theta SR(r_0(\bar{s}_0^\theta, \mu_\theta(\bar{s}_0^\theta), \bar{s}_1^\theta), \dots, r_{T-1}(\bar{s}_{T-1}^\theta, \mu_\theta(\bar{s}_{T-1}^\theta), \bar{s}_T^\theta))], \qquad (3)$$

where $\nabla_\theta SR$ can be obtained by backpropagation using the chain rule. This leads to the following stochastic gradient step:

$$\theta_{k+1} = \theta_k + \alpha \nabla_\theta SR(r_0(\bar{s}_0^\theta, \mu_\theta(\bar{s}_0^\theta), \bar{s}_1^\theta), \dots, r_{T-1}(\bar{s}_{T-1}^\theta, \mu_\theta(\bar{s}_{T-1}^\theta), \bar{s}_T^\theta),$$

with $\alpha > 0$ as the step size.

## 3 THE NEW PERMUTATION INVARIANT WAVECORR ARCHITECTURE

There are several considerations that go into the design of the network for the portfolio policy network $\mu_\theta$. First, the network should have the capacity to handle long historical time series data, which allows for extracting long-term dependencies across time. Second, the network should be flexible in its design for capturing dependencies across a large number of available assets. Third, the network should be parsimoniously parameterized to achieve these objectives without being prone to overfitting. To this end, the WaveNet structure (Oord et al., 2016) offers a good basis for developing our architecture and was employed in Zhang et al. (2020). Unfortunately, a direct application of WaveNet in portfolio management struggles at processing the cross-asset correlation information. This is because the convolutions embedded in the WaveNet model are 1D and extending to 2D convolutions increases the number of parameters in the model, which makes it more prone to the issue of over-fitting, a notorious issue particulary in RL. Most importantly, naive attempts at adapting WaveNet to account for such dependencies (as done in Zhang et al. (2020)) can make the network become sensitive to the ordering of the assets in the input data, an issue that we will revisit below.

We first present the general architecture of WaveCorr in Figure 2. Here, the network takes as input

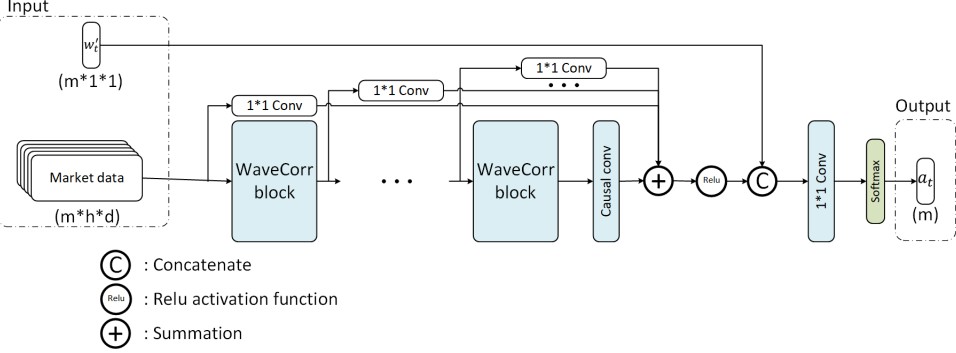

Figure 2: The architecture of the WaveCorr policy network

a tensor of dimension $m \times h \times d$, where $m$ : the number of assets, $h$ : the size of look-back time window, $d$ : the number of channels (number of features for each asset), and generates as output an $m$-dimensional wealth allocation vector. The WaveCorr blocks, which play the key role for extracting cross time/asset dependencies, form the body of the architecture. In order to provide more flexibility for the choice of $h$, we define a causal convolution after the sequence of WaveCorr blocks to adjust the receptive field so that it includes the whole length of the input time series. Also, similar to the WaveNet structure, we use skip connections in our architecture.

The design of the WaveCorr residual block in WaveCorr extends a simplified variation (Bai et al., 2018) of the residual block in WaveNet by adding our new correlation layers (and Relu, concatenation operations following right after). As shown in Figure 3, the block includes two layers of dilated causal convolutions followed by Relu activation functions and dropout layers. Having an input of

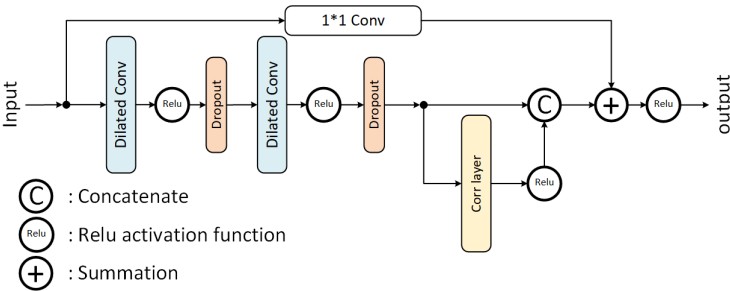

Figure 3: WaveCorr residual block

dimensions $m \times h \times d$, the convolutions output tensors of dimension $m \times h \times d'$ where each slice of the output tensor, i.e. an $m \times 1 \times d$ matrix, contains the dependency information of each asset over time. By applying different dilation rates in each WaveCorr block, the model is able of extracting the dependency information for a longer time horizon. A dropout layer with a rate of $50\%$ is considered to prevent over-fitting, whereas for the gradient explosion/vanishing prevention mechanism of residual connection we use a $1 \times 1$ convolution (presented on the top of Figure 3), which inherently ensures that the summation operation is over tensors of the same shape. The *Corr* layer generates an output tensor of dimensions $m \times h \times 1$ from an $m \times h \times d$ input, where each slice of the output tensor, i.e. an $m \times 1 \times 1$ matrix, is meant to contain cross-asset dependency information. The concatenation operator combines the cross-asset dependency information obtained from the *Corr* layer with the cross-time dependency information obtained from the causal convolutions.

Before defining the *Corr* layer , we take a pause to introduce a property that will be used to further guide its design, namely the property of asset permutation invariance (API). This property is motivated by the idea that the set of possible investment policies that can be modeled by the portfolio policy network should not be affected by the way the assets are indexed in the problem. On a block per block level, we will therefore impose that, when the asset indexing of the input tensor is reordered, the set of possible mappings obtained should also only differ in its asset indexing. More specifically, we let $\sigma : \mathbb{R}^{m \times h \times d} \to \mathbb{R}^{m \times h \times d}$ denote a first coordinate permutation operator over a tensor $\mathcal{T}$ such that $\sigma(\mathcal{T})[i,:,:] = \mathcal{T}[\pi(i),:,:]$, where $\pi : \{1,...,m\} \to \{1,...,m\}$ is a bijective function. Furthermore, we consider $\sigma^{-1} : \mathbb{R}^{m \times h \times d'} \to \mathbb{R}^{m \times h \times d'}$ denote its "inverse" such that $\sigma^{-1}(\mathcal{O})[i,:,:] := \mathcal{O}[\pi^{-1}(i),:,:]$, with $\mathcal{O} \in \mathbb{R}^{m \times h \times d'}$.

**Definition 3.1.** *(Permutation Invariance) A neural network architecture capturing a set of functions* $\mathcal{B} \subseteq \{B : \mathbb{R}^{m \times h \times d} \to \mathbb{R}^{m \times h' \times d'}\}$ *is **permutation invariant** in the first coordinate if given any permutation operator* $\sigma$*, we have that* $\{\sigma^{-1} \circ B \circ \sigma : B \in \mathcal{B}\} = \mathcal{B}$*, where* $\circ$ *stands for function composition. In the context of portfolio policy networks, we refer to the property of permutation invariance in the first coordinate as asset permutation invariance.*

One can verify, for instances, that the architectures of all the neural network blocks described so far in WaveCorr are permutation invariant in the first coordinate, and furthermore that this property is preserved under composition (see Appendix A.2.2).

With this property in mind, we now detail the design of a permutation invariant *Corr* layer via Procedure 1, where we denote as $CC : \mathbb{R}^{(m+1) \times h \times d} \to \mathbb{R}^{1 \times h \times 1}$ the operator that applies an $(m+1) \times 1$ convolution, and as $Concat_1$ the operator that concatenates two tensors along the first dimension. In Procedure 1, the kernel is applied to a tensor in $\mathbb{R}^{(m+1) \times h \times d}$ constructed from adding the $i$-th row of the input tensor to the top of the tensor. Concatenating the output tensors from each run gives the final output tensor. Figure 4 gives an example for the case with $m = 5$, and $h = d = 1$. Effectively, one can show that *Corr* layer satisfies permutation invariance (proof in Appendix).

**Proposition 3.1.** *The Corr layer block satisfies permutation invariance in the first coordinate.*

Table 1 summarizes the details of each layer involved in the WaveCorr architecture: including kernel sizes, internal numbers of channels, dilation rates, and types of activation functions. Overall, the following proposition confirms that this WaveCorr portfolio policy network satisfies asset permutation invariance (see Appendix for proof).

**Proposition 3.2.** *The WaveCorr portfolio policy network architecture satisfies the API property.*

---

**Procedure 1:** *Corr* layer

---

**Result:** Tensor that contains correlation information, $\mathcal{O}_{out}$ of dimension $m \times h \times 1$
**Inputs**: Tensor $\mathcal{O}_{in}$ of dimension $m \times h \times d$;
Define an empty tensor $\mathcal{O}_{out}$ of dimension $0 \times h \times 1$;
**for** $i = 1 : m$ **do**
$\quad$ Set $\mathcal{O}_{out} = Concat_1(\mathcal{O}_{out}, CC(Concat_1(\mathcal{O}_{in}[i,:,:], \mathcal{O}_{in}))))$;
**end**

---

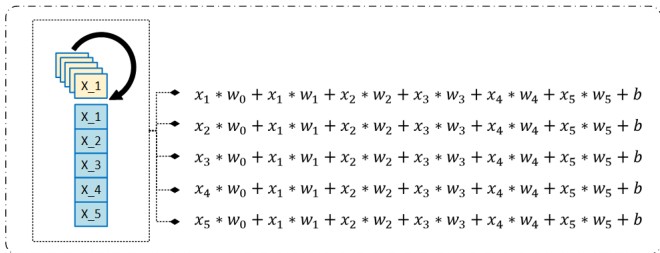

$$x_1 * w_0 + x_1 * w_1 + x_2 * w_2 + x_3 * w_3 + x_4 * w_4 + x_5 * w_5 + b$$
$$x_2 * w_0 + x_1 * w_1 + x_2 * w_2 + x_3 * w_3 + x_4 * w_4 + x_5 * w_5 + b$$
$$x_3 * w_0 + x_1 * w_1 + x_2 * w_2 + x_3 * w_3 + x_4 * w_4 + x_5 * w_5 + b$$
$$x_4 * w_0 + x_1 * w_1 + x_2 * w_2 + x_3 * w_3 + x_4 * w_4 + x_5 * w_5 + b$$
$$x_5 * w_0 + x_1 * w_1 + x_2 * w_2 + x_3 * w_3 + x_4 * w_4 + x_5 * w_5 + b$$

Figure 4: An example of the *Corr* layer over 5 assets

Table 1: The structure of the network

| Layer | Input shape | Output shape | Kernel | Activation | Dilation rate |
|---|---|---|---|---|---|
| Dilated conv | $(m \times h \times d)$ | $(m \times h \times 8)$ | $(1 \times 3)$ | Relu | 1 |
| Dilated conv | $(m \times h \times 8)$ | $(m \times h \times 8)$ | $(1 \times 3)$ | Relu | 1 |
| *Corr* layer | $(m \times h \times 8)$ | $(m \times h \times 1)$ | $([m+1] \times 1)$ | Relu | - |
| Dilated conv | $(m \times h \times 9)$ | $(m \times h \times 16)$ | $(1 \times 3)$ | Relu | 2 |
| Dilated conv | $(m \times h \times 16)$ | $(m \times h \times 16)$ | $(1 \times 3)$ | Relu | 2 |
| *Corr* layer | $(m \times h \times 16)$ | $(m \times h \times 1)$ | $([m+1] \times 1)$ | Relu | - |
| Dilated conv | $(m \times h \times 17)$ | $(m \times h \times 16)$ | $(1 \times 3)$ | Relu | 4 |
| Dilated conv | $(m \times h \times 16)$ | $(m \times h \times 16)$ | $(1 \times 3)$ | Relu | 4 |
| *Corr* layer | $(m \times h \times 16)$ | $(m \times h \times 1)$ | $([m+1] \times 1)$ | Relu | - |
| Causal conv | $(m \times h \times 17)$ | $(m \times 1 \times 16)$ | $(1 \times [h-28])$ | Relu | - |
| $1 \times 1$ conv | $(m \times 1 \times 17)$ | $(m \times 1 \times 1)$ | $(1 \times 1)$ | Softmax | - |

Finally, it is necessary to discuss some connections with the recent work of Zhang et al. (2020), where the authors propose an architecture that also takes both sequential and cross-asset dependency into consideration. Their proposed architecture, from a high level perspective, is more complex than ours in that theirs involves two sub-networks, one LSTM and one CNN, whereas ours is built solely on CNN. Our architecture is thus simpler to implement, less susceptible to overfitting, and allows for more efficient computation. The most noticeable difference between their design and ours is at the level of the *Corr* layer block, where they use a convolution with a $m \times 1$ kernel to extract dependency across assets and apply a standard padding trick to keep the output tensor invariant in size. Their approach suffers from two issues (see Appendix A.3 for details): first, the kernel in their design may capture only partial dependency information, and second, most problematically, their design is not asset permutation invariant and thus the performance of their network can be highly sensitive to the ordering of assets. This second issue is further confirmed empirically in section 4.3.

**Remark 3.1.** *To the best of our knowledge, all the previous literature on permutation invariance (PI) (see for e.g. Zaheer et al. (2017); Cai et al. (2021); Li et al. (2021)) of neural networks have considered a definition that requires (see Definition 1 in Cai et al. (2021)) that all functions $B \in \mathcal{B}$ are PI, i.e. that for any permutation operator $\sigma$, we have that $\sigma^{-1} \circ B \circ \sigma = B$. Definition 3.1 is more flexible as it focuses on the representation power of the NN architecture rather than the resulting NN, i.e. it does not impose the PI property on every functions of the set $\mathcal{B}$ but rather on the set as a whole, i.e. if $B \in \mathcal{B}$ then $\sigma^{-1} \circ B \circ \sigma$ is also in $\mathcal{B}$ for all $\sigma$. This distinction is crucial given that Cai et al. (2021) observe that correlations cannot be modeled using their definition of PI networks "as they are agnostic to identities of entities". In fact, the architecture proposed in Figure 3 violates Cai et al.'s PI property and ends up not suffering from this deficiency.*

## 4 EXPERIMENTAL RESULTS

In this section, we present the results of a series of experiments evaluating the empirical performance of our WaveCorr DRL approach. We start by presenting the experimental set-up. We follow with our main study that evaluates WaveCorr against a number of popular benchmarks. We finally shed light on the superior performance of WaveCorr with comparative studies that evaluate the sensitivity of its performance to permutation of the assets, number of assets, size of transaction costs, and (in Appendix A.6.3) maximum holding constraints. All codes are available at https://anonymous.4open.science/r/waveCorr-7748.

### 4.1 EXPERIMENTAL SET-UP

**Data sets:** We employ three data sets. **Can-data** includes the daily closing prices of 50 Canadian assets from 01/01/2003 until 01/11/2019 randomly chosen among the 70 companies that were continuously part of the Canadian S&P/TSX Composite Index during this period. **US-data** contains 50 randomly picked US assets among the 250 that were part of S&P500 index during the same period. Finally, **Covid-data** considered 50 randomly resampled assets from S&P/TSX Composite Index for period 01/11/2011-01/01/2021 and included open, highest, lowest, and closing daily prices. The Can-data and US-data sets were partitioned into training, validation, and test sets according to the periods 2003-2009/2010-2013/2014-2019 and 2003-2009/2010-2012/2013-2019 respectively, while the Covid-data was only divided in a training (2012-2018) and testing (2019-2020) periods given that hyper-parameters were reused from the previous two studies. We assume with all datasets a constant comission rate of $c_s = c_p = 0.05\%$ in the comparative study, while the sensitivity analysis considers no transaction costs unless specified otherwise.

**Benchmarks:** In our main study, we compare the performance of WaveCorr to CS-PPN (Zhang et al., 2020), EIIE (Jiang et al., 2017), and the equal weighted portfolio (EW). Note that both CS-PPN and EIIE were adapted to optimize the Sharpe-ratio objective described in section 2.2 that exactly accounts for transaction costs.

**Hyper-parameter selection:** Based on a preliminary unreported investigation, where we explored the influence of different optimizers (namely ADAM, SGD, RMSProp, and SGD with momentum), we concluded that ADAM had the fastest convergence. We also narrowed down a list of reasonable values (see Table A.6) for the following common hyper-parameters: initial learning rate, decay rate, minimum rate, look-back window size $h$, planning horizon $T$. For each method, the final choice of hyper-parameter settings was done based on the average annual return achieved on both a 4-fold cross-validation study using Can-data and a 3-fold study with the US-data. The final selection (see Table A.6) favored, for each method, a candidate that appeared in the top 5 best performing settings of both data-sets in order to encourage generalization power among similarly performing candidates. Note that in order to decide on the number of epochs, an early stopping criteria was systematically employed.

**Metrics:** We evaluated all approaches using out-of-sample data ("test data"). "Annual return" denotes the annualized rate of return for the accumulated portfolio value. "Annual vol" denotes the prorated standard deviation of daily returns. Trajectory-wise Sharpe ratio (SR) of the log returns, Maximum drawdown (MDD), i.e. biggest loss from a peak, and average Turnover, i.e. average of the trading volume, are also reported (see (Zhang et al., 2020) for formal definitions). Finaly, we report on the average "daily hit rate" which captures the proportion of days during which the log returns out-performed EW.

**Important implementation details:** Exploiting the fact that our SGD step involves exercising the portfolio policy network for $T$ consecutive steps (see equation (3)), a clever implementation was able to reduce WavCorr's training time per episode by a factor of 4. This was done by replacing the $T$ copies of the portfolio policy network producing $a_0, a_2, \ldots, a_{T-1}$, with an equivalent single augmented multi-period portfolio policy network producing all of these actions simultaneously, while making sure that all intermediate calculations are reused as much as possible (see Appendix A.4 for details). We also implemented our stochastic gradient descent approach by updating, after each episode $k$, the initial state distribution $F$ to reflect the latest policy $\mu_{\theta_k}$. This was done in order for

the final policy to be better adapted to the conditions encountered when the portfolio policy network is applied on a longer horizon than $T$.

## 4.2 COMPARATIVE EVALUATION OF WAVECORR

In this set of experiments the performances of WaveCorr, CS-PPN, EIIE, and EW are compared for a set of 10 experiments (with random reinitialization of NN parameters) on the three datasets. The average and standard deviations of each performance metric are presented in Table 2 while Figure A.8 (in the Appendix) presents the average out-of-sample portfolio value trajectories. The main takeaway from the table is that WaveCorr significantly outperforms the three benchmarks on all data sets, achieving an absolute improvement in average yearly returns of 3% to 25% compared to the best alternative. It also dominates CS-PPN and EIIE in terms of Sharpe ratio, maximum drawdown, daily hit rate, and turnover. EW does appear to be causing less volatility in the US-data, which leads to a slightly improved SR. Another important observation consists in the variance of these metrics over the 10 experiments. Once again WaveCorr comes out as being generally more reliable than the two other DRL benchmarks in the Can-data, while EIIE appears to be more reliable in the US-data sacrificing average performance. Overall, the impressive performance of WaveCorr seems to support our claim that our new architecture allows for a better identification of the cross-asset dependencies. In conditions of market crisis (i.e. the Covid-data), we finally observe that WaveCorr exposes the investors to much lower short term losses, with an MDD of only 31% compared to more than twice as much for CS-PPN and EIIE, which reflects of a more effective hedging strategy.

Table 2: The average (and standard deviation) performances using three data sets.

| Method | Annual return | Annual vol | SR | MDD | Daily hit rate | Turnover |
|---|---|---|---|---|---|---|
| | | | Can-data | | | |
| WaveCorr | 27% (3%) | 16% (1%) | 1.73 (0.25) | 16% (2%) | 52% (1%) | 0.32 (0.01) |
| CS-PPN | 21% (4%) | 19% (2%) | 1.14 (0.34) | 17% (4%) | 51% (1%) | 0.38 (0.05) |
| EIIE | -1% (8%) | 29% (4%) | -0.01 (0.28) | 55% (9%) | 47% (1%) | 0.64 (0.08) |
| EW | 4% (0%) | 14% (0%) | 0.31 (0.00) | 36% (0%) | - | 0.00 (0.00) |
| | | | US-data | | | |
| WaveCorr | 19% (2%) | 16% (2%) | 1.17 (0.20) | 20% (4%) | 50% (1%) | 0.11 (0.02) |
| CS-PPN | 14% (2%) | 15% (2%) | 0.94 (0.17) | 22% (6%) | 49% (1%) | 0.15 (0.08) |
| EIIE | 16% (1%) | 15% (0%) | 1.09 (0.06) | 20% (1%) | 50% (0%) | 0.17 (0.02) |
| EW | 15% (0%) | 13% (0%) | 1.18 (0.00) | 18% (0%) | - | 0.00 (0.00) |
| | | | Covid-data | | | |
| WaveCorr | 56% (13%) | 26% (5%) | 2.16 (0.50) | 31% (9%) | 51% (2%) | 0.19 (0.05) |
| CS-PPN | 31% (27%) | 51% (6%) | 0.60 (0.48) | 67% (7%) | 50% (2%) | 0.3 (0.09) |
| EIIE | 11% (30%) | 76% (17%) | 0.20 (0.43) | 77% (13%) | 46% (2%) | 0.76 (0.27) |
| EW | 27% (0%) | 29% (0%) | 0.93(0.00) | 47% (0%) | - | 0.01 (0.00) |

## 4.3 SENSITIVITY ANALYSIS

**Sensitivity to permutation of the assets:** In this set of experiment, we are interested in measuring the effect of asset permutation on the performance of WaveCorr and CS-PPN. Specifically, each experiment now consists in resampling a permutation of the 50 stocks instead of the initial parameters of the neural networks. The results are summarized in Table 3 and illustrated in Figure 5. We observe that the learning curves and performance of CS-PPN are significantly affected by asset permutation compared to WaveCorr. In particular, one sees that the standard deviation of annual return is reduced by more than a factor of about 5 with WaveCorr. We believe this is entirely attributable to the new PI structure of the *Corr* layer in the portfolio policy network.

Table 3: The average (and standard dev.) performances over random asset permutation in Can-data.

| | Annual return | Annual vol | SR | MDD | Daily hit rate | Turnover |
|---|---|---|---|---|---|---|
| WaveCorr | 48% (1%) | 15% (1%) | 3.15 (0.19) | 14% (3%) | 56% (0%) | 0.48 (0.01) |
| CS-PPN | 35% (5%) | 18% (1%) | 2.00 (0.37) | 22% (4%) | 54% (1%) | 0.54 (0.03) |

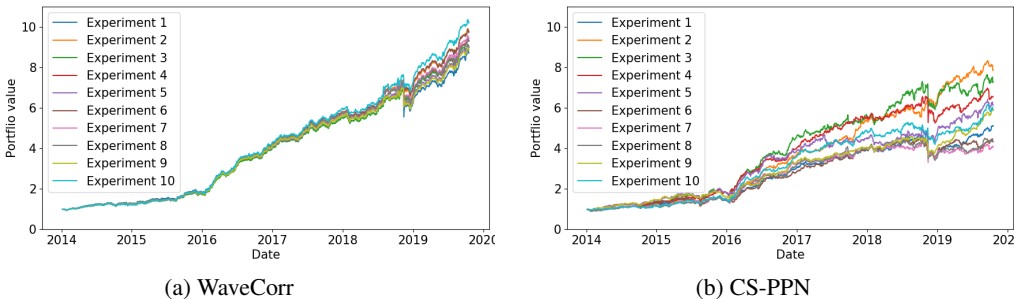

Figure 5: Comparison of the wealth accumulated by WaveCorr and CS-PPN under random initial permutation of assets on Can-data's test set.

**Sensitivity to number of assets:**    In this set of experiments, we measure the effect of varying the number of assets on the performance of WaveCorr and CS-PPN. We therefore run 10 experiments (randomly resampling initial NN parameters) with growing subsets of 30, 40, and 50 assets from Can-data. Results are summarized in Table 4 and illustrated in Figure A.9 (in Appendix). While having access to more assets should in theory be beneficial for the portfolio performance, we observe that it is not necessarily the case for CS-PPN. On the other hand, as the number of assets increase, a significant improvement, with respect to all metrics, is achieved by WaveCorr. This evidence points to a better use of the correlation information in the data by WaveCorr.

Table 4: The average (and std. dev.) performances as a function of the number of assets in Can-data.

| # of stocks | Annual return | Annual vol | SR | MDD | Daily hit rate | Turnover |
|---|---|---|---|---|---|---|
| | | | WaveCorr | | | |
| 30 | 37.7% (4%) | 19% (1%) | 2.02 (0.27) | 22% (3%) | 55% (1%) | 0.39 (0.02) |
| 40 | 38.5% (4%) | 21% (1%) | 1.81 (0.17) | 23% (2%) | 55% (1%) | 0.44 (0.04) |
| 50 | 43.0% (5%) | 17% (2%) | 2.57 (0.52) | 20% (6%) | 55% (1%) | 0.43 (0.02) |
| | | | CS-PPN | | | |
| 30 | 30.3% (3%) | 17% (1%) | 1.80 (0.20) | 21% (4%) | 53% (1%) | 0.42 (0.04) |
| 40 | 29.8% (7%) | 17% (2%) | 1.70 (0.34) | 22% (3%) | 53% (1%) | 0.41 (0.09) |
| 50 | 32.2% (4%) | 16% (1%) | 2.07 (0.28) | 18% (3%) | 52% (1%) | 0.43 (0.05) |

**Sensitivity to commission rate:**    Table 5 presents how the performances of WaveCorr and CS-PPN are affected by the magnitude of the commission rate, ranging among 0% and 0.05%. One can first recognize that the two methods appear to have good control on turnover as the commission rate is increased. Nevertheless, one can confirm from this table the significantly superior performance of WaveCorr prevails under both level of commission rate.

Table 5: The average (and std. dev.) performances as a function of commission rate (CR) in Can-data.

| Method | Annual return | Annual vol | SR | MDD | Daily hit rate | Turnover |
|---|---|---|---|---|---|---|
| | | | CR = 0 | | | |
| WaveCorr | 42% (3%) | 15% (0%) | 2.77 (0.20) | 13% (1%) | 55% (1%) | 0.44 (0.02) |
| CS-PPN | 35% (4%) | 17% (1%) | 2.04 (0.27) | 14% (3%) | 53% (1%) | 0.47 (0.05) |
| | | | CR = 0.05% | | | |
| WaveCorr | 27% (3%) | 16% (1%) | 1.73 (0.25) | 15% (2%) | 52% (1%) | 0.32 (0.01) |
| CS-PPN | 21% (4%) | 19% (2%) | 1.14 (0.34) | 17% (4%) | 51% (1%) | 0.38 (0.05) |

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
