# OpenReview forum: "WaveCorr: Deep Reinforcement Learning with Permutation Invariant Policy Networks for Portfolio Management"
_ICLR.cc/2022/Conference — ICLR 2022 Submitted_

### Official Review · Reviewer_y3fW · 2021-10-17

**Correctness:** 3
**Technical Novelty And Significance:** 3
**Empirical Novelty And Significance:** 3
**Recommendation:** 8
**Confidence:** 3

**Main Review:**

I have following major comments on the paper:

Pros:

- For financial data, the estimation of correlation structure is a notoriously hard problem prone to overfitting, due to low signal-to-noise ratio and high dimensional nature of covariance matrix. In my personal view, the PI principle can be viewed as another regularization structure to reduce the complexity of the function class, thus alleviate the over-fitting problem. The idea of adding the corr layer and the result that **WaveCorr** portfolio policy network architecture satisfies the PI property are interesting.

- The numerical section is well structured and the training procedures are reported in detail.

Cons:
- The paper discusses in Remark 3.1 about the distinction of definition of PI between this paper and the previous literature, but it seems to me that the motivation for this part is still insufficient. Except for the fact that the architecture in this paper violates the PI in previous literature, is there any other motivation or intuition to define PI in the current setting?

- The current approach seems also applicable to long-short portfolio (i.e., $\sum w_i = 0$). Is it possible to include some simple test in the appendix to test the performance in this setting?

- In the numerical part, the standard deviations reported in tables solely reflect the randomness in selection of assets, but fail to capture the uncertainty from temporal variation. For example, one question to consider is: what would be the ex-ante standard deviation estimation, if the proposed portfolio is traded in the next year? If this part is not addressed, one can not rule out the possibility that the superior performance of the proposed method is due to a lucky test period.

**Summary Of The Paper:**

The paper develops the first CNN based portfolio optimization network named **WaveCorr**, which is capable of capturing both temporal and cross-sectional correlation structure for the training data. The paper proposes a new type of the permutation invariant (PI) principle for a class of functions (e.g., neural network blocks), and it demonstrates that \texttt{WaveCorr} structure satisfies the PI principle. By conducting numerical studies using data from both Canadian (TSX) and American (S\&P 500) stock markets, the paper testifies the superior performance of **WaveCorr** in terms of Sharpe ratio and stability, compared with several benchmark models.

**Summary Of The Review:**

In general the paper is well-written, the motivation is clear, and the numerical result is convincing. I recommend acceptance for this paper.

---

> ### Author Response · Authors · 2021-11-19
> **Response to y3fW**
>
> We thank the reviewer for his/her response. Here is our response to the concerns that are raised.
>
> Regarding “The paper discusses in Remark 3.1...”, the paper currently mentions two motivations for this property. 1) P.5 “This property is motivated by the idea that the set of possible investment policies that can be modeled by the portfolio policy network should not be affected by the way the assets are indexed in the problem.” and 2) P.6 “correlations cannot be modeled using [Cai et al.’s] definition of PI networks “as they are agnostic to identities of entities””. In designing the portfolio policy network, we therefore needed an alternate definition of PI property and thus designed a property that needs to be verified on the NN architecture as a whole instead of being verified for every mapping produced by the architecture. This seems to be the most natural relaxation of Cai et al’s PI property, which is the only version of PI property we found in the deep learning literature.
>
> Regarding “The current approach seems also applicable to long-short portfolio”, we agree with the reviewer that our approach could accommodate long-short portfolios with minor changes. We decided to keep the paper focused on long only positions due to the popularity of this setting in the literature.
>
> Regarding “In the numerical part, the standard deviations...”, We would like to bring the reviewer’s attention to the fact that while the test set of Can-data and US-data both cover 2014-2019, the test set of Covid-data did cover the different period of 2019-2020, which includes noticeably different market behaviors. While we agree that it would be necessary to see additional results from other data sets and time periods before deploying WaveCorr in practice, we believe that our experiment design is more than sufficient to support our claim that the WaveCorr architecture improves on CS-PPN and EIIE because (respectively) of its PI property and its aptitude to exploit correlations.

---

### Official Review · Reviewer_iRgh · 2021-11-02

**Correctness:** 3
**Technical Novelty And Significance:** 2
**Empirical Novelty And Significance:** 2
**Recommendation:** 5
**Confidence:** 2

**Main Review:**

Strength
-  Applying deep reinforcement learning to address the portfolio management problem is an interesting yet less explored direction in the RL community.
- The proposed permutation invariant (equivariance) architecture seems effective. It outperforms baselines in multiple datasets.

Weakness
   - The reviewer has some questions / concerns about the clarity of some technical details. Please see the comments below:
	    - It is unclear how to obtain the random vector of asset returns. Is it estimated from the historical data? How do you construct the random vector?
	    - To achieve permutation invariance, graph neural networks and self-attention mechanisms are widely used in the literature. What is the motivation for using Corr layer instead of popular methods such as graph neural net or transformer?
	    - The details for the RL algorithm are missing. What is the RL algorithm used to optimize the policy. Section 2.2 describes a vanilla policy gradient. Is vanilla policy gradient used in the experiments? In addition, the hyper-parameters used in the RL algorithms should be provided.
	  - Remark 3.1 confuses the reviewer. Specifically, it reads “the architecture proposed in Figure 3 violates Cai et al.’s PI property”. What is Cai et al.’s PI property? What’s the difference between Cai et al.’s PI property and that defined in Definition 3.1?
- Experiments
   - Some metrics seem not defined. For instance, the definition of “Maximum drawdown”, “Daily hit rate”, and “turnover” are missing.
   - How many environment steps are used to train the policy?



**Summary Of The Paper:**

This paper applies deep reinforcement learning to address the problem of portfolio management where decisions need to be made over time and many factors such as market conditions need to be considered.
This paper proposed a portfolio policy network that has the permutation invariance (equivariance) property when treating multiple assets’ information. The proposed approach is evaluated on three sets of data from the Canadian and US stock markets. The results show that the proposed approach outperformed baselines in terms of different metrics such as annual return and Sharpe ratio.


**Summary Of The Review:**

In summary, the reviewer found applying deep RL to portfolio management interesting. However, the reviewer has some concern about the clarity of the presentation. Clarification for some technical contents are needed.

---

> ### Author Response · Authors · 2021-11-19
> **Response to iRgh**
>
> We thank the reviewer for his/her response. Here is our response to the concerns that are raised.
>
> Regarding “It is unclear how to obtain the random vector of asset returns…” the window of asset returns is built based on historical asset prices obtained from our financial data source.
>
> Regarding “To achieve permutation invariance, graph neural networks and self-attention mechanisms are widely used”. The reviewer is right that a graph neural network or set transformer as in (Lee et al. 2019) would satisfy our permutation invariance property. Yet, those neural network architectures are actually designed to satisfy the stronger definition of permutation invariance that we refer to as Cai et al.’s PI property in Remark 3.1. Remark 3.1 argues that this property is too strict in the context of treating asset entities as it prevents the neural network from learning to exploit correlations between assets present in the training data. To put it in Cai et al’s own words, networks that satisfy Cai et al.’s PI property “are agnostic to identities of entities” they can therefore not learn that for e.g. stock #1 is correlated to stock #2.
>
> Regarding “The details for the RL algorithm are missing ...”, we apologize for this confusion. The RL algorithm used in this work is the standard actor-only direct policy gradient method (Baxter and Bartlett (2000)), which has been applied for example in (Moody and Saffell (2001), Moody et al. (1998)) under the name of Direct Reinforcement, and (Zhang et al. (2020), Jiang et al. (2017)) under the name of Direct Policy Gradient (DPG). Finally, regarding the hyperparameters, we would like to bring the reviewer’s attention to Table A.6 in the appendix.
>
>
> Regarding “ Remark 3.1 confuses the reviewer….”, we are sorry to read that the reviewer is confused with remark 3.1 since it plays the important role of explaining how our PI definition is different from the traditional definition (that we call Cai et al’ PI property). We are however happy to help the reviewer distinguish between the two properties. First, we need to emphasize that they are fundamentally different. In particular, as mentioned in Remark 3.1, Cai et al’s PI property imposes a condition on every mapping that can be modeled by the network architecture. This is what we describe mathematically as $\forall B \in \mathcal{B},\forall\sigma$, $\sigma^{-1}\circ B\circ \sigma = B$. Our PI definition is different as it only imposes a condition on the whole architecture and not specifically on every mapping that it can model. Specifically, for any mapping that can be produced, and every permutation plan, there must be another mapping in the range of the network that is equivalent to the first one when the indexes of the input matrix are permuted. Mathematically, $B\in\mathcal{B} \Rightarrow \forall \sigma,\; \sigma^{-1}\circ B\circ \sigma\in \mathcal{B}$. In particular, architectures that are PI with respect to Cai et al.’s definition are also PI with respect to ours. Yet, the reverse is not true as described in the next example.
>
> To give a simple example that explains the differences, Cai et al.’s definition would not consider a fully connected neural network architecture to be permutation invariant since once it is trained, it will usually happen that permuting indexes in the input matrix will generate a different output from the NN. In comparison, our definition does consider the fully connected network architecture permutation invariant. The reason is that any mapping that can be obtained from the NN architecture (after training) for a given input matrix indexing definition, can just as well be obtained from the same architecture if the input matrix indexing is defined differently and the network is retrained under this new indexing definition.
>
> Regarding “Some metrics seem not defined…”, we will be happy to include later an appendix that presents these definitions.
>
> Regarding “How many environment steps are used to train the policy” the answer is 5000 as mentioned in Table A.6, list of hyper-parameters.
>
> References:
>
> Lee, J., Lee, Y., Kim, J., Kosiorek, A. R., Choi, S., Teh, Y. W., Set Transformer: A Framework for Attention-based Permutation-Invariant Neural Networks, ICML, 2019.
>
> Baxter, J., and Bartlett, P. L.. "Direct gradient-based reinforcement learning." 2000 IEEE International Symposium on Circuits and Systems (ISCAS). Vol. 3. IEEE, 2000.
>
> Moody, J., Wu, L., Liao, Y., & Saffell, M. (1998). Performance functions and reinforcement learning for trading systems and portfolios. Journal of Forecasting, 17(5‐6), 441-470.
>
> Moody, J., & Saffell, M. (2001). Learning to trade via direct reinforcement. IEEE transactions on neural Networks, 12(4), 875-889.

---

> > ### Comment · Reviewer_iRgh · 2021-11-29
> > **Thanks for the Response**
> >
> > Thanks for the response. After reading the response, the reviewer still has concerns about Definition 3.1. It is unclear to the reviewer what's the main difference between the proposed "PI property" and the "permutation equivariance" property [a, b], which can be achieved by graph nets with parameter sharing and is widely studied by many previous works such as [a, b].
> >
> > [a] Equivariance Through Parameter-Sharing, Ravanbakhsh, ICML'17
> > [b] Invariant and Equivariant Graph Networks, Maron, ICLR'19

---

### Official Review · Reviewer_4KRB · 2021-11-05

**Correctness:** 4
**Technical Novelty And Significance:** 2
**Empirical Novelty And Significance:** Not applicable
**Recommendation:** 5
**Confidence:** 3

**Main Review:**

Strength:

1. The paper presents a clear formulation of the portfolio optimization problem in section 2. The illustration of the architecture and its specificity is also detailed.

2. The paper presents a reasonable solution to deal with permutation invariance across assets, which is a relevant problem in portfolio optimization with deep learning.

3. The paper presents experimental results on real-world data that demonstrate the effectiveness of the proposed method compared to the alternatives in consideration.

Weakness:
1. Presentation: The description of the architecture is not clear. The intention of some of the components, such as causal convolution, skip connections are not specified. Related work such as WaveNet is not well discussed.

2. Contribution: While the paper identifies the asset permutation invariance property and proposed a solution to deal with this problem in portfolio optimization, I find this contribution somewhat limited because permutation invariance as a general property is not new.

3. Experiments: While the proposed method is validated on various datasets competing with various existing methods, it is not clear to me whether the compared methods are indeed state-of-the-art as claimed. It is also not clear to me whether the experiments indeed demonstrate that the proposed method can capture cross-asset information.

Question:
The paper considers permutation across assets m, is it necessary to also consider permutation across channels d?

**Summary Of The Paper:**

The paper presents a deep reinforcement learning model for portfolio optimization that harnesses both cross-asset dependencies as well as time dependencies. To achieve this goal, the paper presents a residue block that is consists of dilated convolution component for time dependency and Correlation component for cross-asset dependency.  Experiments are carried out on three datasets using price data and compared to three existing methods to demonstrate the effectiveness of the proposed method.

**Summary Of The Review:**

Correctness: I did not spot any obvious errors in the paper.
Novelty and significance: this is my major concern. It is unclear to me whether the proposed method is original and significant enough.

---

> ### Author Response · Authors · 2021-11-19
> **Response to 4KRB**
>
> We thank the reviewer for his/her response. Here is our response to the concerns that are raised:
>
> Regarding Weakness #1: As mentioned in the paper, the main architecture of the paper is based on the well-known WaveNet. (Oord et al., 2016) carefully elaborates on every element and the purpose of using them. In particular, they explain that the reason for embedding residual networks in their architecture is to avoid gradient vanishing and explosion. Note that we do mention in our paper at the top of page 5: “A dropout layer with a rate of 50% is considered to prevent overfitting, whereas for the gradient explosion/vanishing prevention mechanism of residual connection we use a 1 * 1 convolution”. In addition, regarding the causal convolution layer we do provide some motivation by mentioning “we define a causal convolution after the sequence of WaveCorr blocks to adjust the receptive field so that it includes the whole length of the input time series” thus clarifying that this layer serves the purpose of taking the whole length of input time series window into account when computing the output. Overall, the explanation of the architecture is brief due to page limit but follows the norm in the literature that includes describing the sequence of different layers, their activation functions, number of channels, kernel size, etc..
>
> Regarding Weakness #2: We wish to refute the underlying claim that "permutation invariance" as defined in our Definition 3.1 has already been studied in ML. To the best of our knowledge (and we encourage the reviewer to provide us a reference if he thinks we are wrong), all the previous literature on permutation invariance (PI) of neural networks have considered a definition that resembles Definition 1 in Cai et al. (see remark 3.1 for detailed distinction in definitions). In fact, Cai et al’s definition is analogous to the property of rotation or translation invariance in image recognition. Our definition is fundamentally different to the earlier definitions of PI.
>
> To give a simple example that explains the differences, Cai et al.’s definition would not consider a fully connected neural network architecture to be permutation invariant since once it is trained, it will usually happen that permuting indexes in the input matrix will generate a different output from the NN. In comparison, our definition does consider the fully connected network architecture permutation invariant. The reason is that any mapping that can be obtained from the NN architecture (after training) for a given input matrix indexing definition, can just as well be obtained from the same architecture if the input matrix indexing is defined differently and the network is retrained under this new indexing definition.
>
> Regarding Weakness #3: We wish to emphasize that we consider the compared methods, i.e.CS-PPN (Zhang et al., 2020), EIIE (Jiang et al., 2017), as the state-of-the-art "deep RL architectures" for solving dynamic portfolio management problems. This claim is based on our extensive search of the literature on the application of deep RL to portfolio management, and we find that these two are the latest and most cited ones. It appears to us that the development of deep RL to solve challenging dynamic portfolio management problems is still at its early stage and we believe our work contributes to identifying a fundamental issue that is particularly important in the design of a "portfolio" policy network and providing a novel solution to resolve the issue. If the reviewer has any other references on deep RL for solving dynamic portfolio management problems, we would be happy to know.
>
> The effectiveness of WaveCorr to capture cross-asset information is demonstrated by benchmarking against portfolio networks that do not take into account cross-asset correlation, i.e. EIIE (Jiang et al. 2017), and the networks that take into account cross-asset information without satisfying our PI property, i.e. CS-PPN (Zhang et al. 2020). As a portfolio policy that exploits cross-asset information can only achieve better performance when there is correlation among assets, the superior performance of WaveCorr against other portfolio networks indicates the effectiveness of WaveCorr to capture cross-asset information.
>
> Regarding the question “The paper considers permutation across assets m, is it necessary to also consider permutation across channels d”, we wish to mention that the architecture that we employ is permutation invariant across channels given that the kernels have the same number of channels as the input tensor. This can be shown to imply that WaveCorr satisfies our definition of permutation invariance with respect to channels. We however prefer not to mention this so that our paper focuses on the motivation and validation of the permutation invariance property with respect to asset indexing.

---

### Official Review · Reviewer_ruHv · 2021-11-07

**Correctness:** 3
**Technical Novelty And Significance:** 3
**Empirical Novelty And Significance:** 3
**Recommendation:** 8
**Confidence:** 3

**Main Review:**

Strengths:
1. The target issue of creating permutation invariant policy networks for DRL is fundamentally interesting and important.
2. The proposed design is well-grounded and has potential applications.
3. The theoretical characterizations are appreciated.

Weaknesses:
1. The portfolio management task is a good example but should be treated as an application, not a goal.
    Such a permutation invariant policy network may have more applications.
2. For comparison methods of the portfolio management task, beside equal weight (EW), it is also expected to compare with minimum variance or mean-variance strategy, which are also widely used.
3. In Table 2~4, the maximum drawdown seems to be relatively high, over 20%?  which is not normal. Please check carefully.



**Summary Of The Paper:**

In this paper, the authors propose a new portfolio policy network architecture for DRl to exploit cross-asset dependency information, which is shown to be able to achieve better performance than existing ones. Such a scheme mainly introduces a permutation invariance property, and it is very interesting.

**Summary Of The Review:**

Interesting design with potential applications.

---

> ### Author Response · Authors · 2021-11-19
> **Response to ruHv**
>
> We thank the reviewer for his/her response. Here is our response to the concerns that are raised:
>
> Regarding Weakness #1: We agree with the reviewer that permutation invariance could benefit deep learning policy networks in other domains. However, we chose portfolio management, which is among the most challenging sequential decision making problems due to the structure of time series data. In particular, the issue of overfitting is notoriously difficult in financial data, which is known to be inherently noisy and non-stationary (see e.g. [1,2]). Hence, proposing a network architecture that is specifically designed with our PI property in mind and can handle time series data in a portfolio management problem demonstrates the added value of using a PI architecture.
>
> Regarding Weakness #2: We need to clarify here that our EW benchmark is actually an EW rebalanced daily and has been used as the benchmark in many other portfolio investment studies where it is considered very difficult to beat [3]. In terms of comparing our model with benchmarks other than EW, we would like to indicate to the reviewer that Zhang et al. (2020) already performed extensive experiments demonstrating that CS-PPN outperformed a large number of popular benchmarks. We therefore decided to use CS-PPN as our main benchmark.
>
> Regarding Weakness #3: The high maximum drawdown that is observable in these tables is due to the fact that the portfolio policy considered in this paper only includes long positions, hence, at the times of big market crashes no hedging strategy can efficiently prevent high values of drawdown. As we can see in these tables, the EW approach, which represents the average of the market, experiences similar high drawdown periods.
>
> [1] Abu-Mostafa, Y. S., & Atiya, A. F. (1996). Introduction to financial forecasting. Applied intelligence, 6(3), 205-213.
>
> [2] Tay, F. E., & Cao, L. (2001). Application of support vector machines in financial time series forecasting. omega, 29(4), 309-317.
>
> [3] DeMiguel, V., Garlappi, L., Uppal, R., Optimal Versus Naive Diversification: How Inefficient is the 1/N Portfolio Strategy?, The Review of Financial Studies, 22(5), 1915–1953, 2009.

---

### Comment · Area_Chair_aLgb · 2021-11-20
**Please read responses from the authors**

Dear reviewers,

Please read the detailed responses from the authors. How do they change your evaluation? Do you still have major concerns? Thank you.

---

### Decision · Program_Chairs · 2022-01-20

**Decision:**

Reject

**Comment:**

This paper proposes an architecture of a policy network (WaveCorr) that is particularly effective for portfolio management tasks.  A key observation that leads to the design of WaveCorr is that the dependency across asset should be treated differently from the dependency across time.  The proposed WaveCorr has the property that it is "permutation invariant" with respect to assets, which means that the class of functions that can be represented by WaveCorr is invariant to permutation of assets.  WaveCorr is shown to achieve the state-of-the-art performance in a portfolio management task.

A major point of discussion was the definition of "permutation invariance".  The reviewers and AC understood the difference between the permutation invariance defined in this paper and that studied in the prior work (the output of a network is insensitive to the permutation of the particular values of the input).  With the definition in this paper, however, a fully connected layer is permutation invariant, but the Corr layer proposed in the paper appears to have more structure.  It is unclear exactly what properties of the Corr layer leads to the performance improvement.